# Genomic analysis of cardiac surgery-associated *Mycobacterium chimaera* infections in Italy

Arash Ghodousi[1], Emanuele Borroni[1], Marta Peracchi[2], Giorgio Palù[3], Loredana Fallico[4], Mario Rassu[4], Vinicio Manfrin[4], Paola Mantegani[1], Vincenzina Monzillo[5], Riccardo Manganelli[3], Enrico Tortoli[1], Daniela Maria Cirillo[1]*

**1** Emerging Bacterial Pathogens Unit, Division of Immunology, Transplantation and Infectious Diseases, IRCCS San Raffaele Scientific Institute, Milan, Italy, **2** Azienda Ospedaliera di Padova, Padova, Italy, **3** University of Padova, Padova, Italy, **4** San Bortolo Hospital, Vicenza, Italy, **5** U.O.C Microbiologia e Virologia, Fondazione IRCCS Policlinico San Matteo, Pavia, Italy

\* cirillo.daniela@hsr.it

**Data Availability Statement:** All sequence reads were submitted to the NCBI Sequence Read Archive (Project number PRJNA592124).

## Abstract

One hundred and twenty-two *Mycobacterium chimaera* strains isolated in Italy from cardiac surgery-related patients, cardiac surgery-unrelated patients and from heater-cooler units, were submitted to whole-genome sequencing and to subsequent SNP analysis. All but one strains isolated from cardiac surgery-related patients belonged to Subgroup 1.1 (19/23) or Subgroup 1.8 (3/23). Only 28 out of 79 strains isolated from heater-cooler units belonged to groupings other than 1.1 and 1.8. The strains isolated from cardiac surgery-unrelated patients were instead distributed across the phylogenetic tree. Our data, the first on isolates from Italy, are in agreement with a recent large genomic study suggesting a common source, represented by strains belonging to Subgroups 1.1 and 1.8, of cardiac surgery-related *Mycobacterium chimaera* infections. The strains belonging to groupings other than 1.1 and 1.8 isolated from heather-cooler units evidently resulted from contaminations at hospital level and had no share in the *Mycobacterium chimaera* outbreak. One *Mycobacterium chimaera* strain investigated in this study proved distant from every previously known *Mycobacterium chimaera* Groups (1, 2, 3 and 4) and we propose to assign to a novel group, named "Group 5".

## Introduction

A global outbreak of *Mycobacterium chimaera* (*M. chimaera*) infections associated with open heart cardiac surgery is ongoing. Since 2013, when the first cases, dated back to 2011, were discovered [1] more than 140 cases of severe *M. chimaera* infection have been identified worldwide in patients who had undergone cardiothoracic surgery with extracorporeal circulation [2]. Very early it emerged that the specific heater-cooler units (HCUs) used in the operatory rooms were contaminated by *M. chimaera* and likely represented the source of infection [3]. Whole genome sequencing (WGS) of *M. chimaera* strains isolated from patients and HCUs of

**Funding:** The authors received no specific funding for this work.

**Competing interests:** The authors have declared that no competing interests exist.

a specific brand and model (Livanova 3T, Germany), both in hospitals and at the factory, has revealed high level of genetic similarity making the most plausible hypothesis a point source contamination of the devices during manufacture [4].

Very little is known about the relation of the Italian clinical and environmental isolates to the global epidemic.

We report here the results of WGS analysis conducted on 122 *M. chimaera* isolates from cardiac surgery (CS) related and unrelated patients and from HCUs in different centres in Italy.

## Materials and methods

A total of 122 Italian isolates of *M. chimaera* were included in this study; 79 from HCUs and 43 from patients, collected between 2015–2019. All clinical *M. chimaera* isolates were received from different hospitals together with medical records of the patients. All the isolates and the patients' medical reports were totally anonymized before our access. A written informed consent to publish the data was obtained from all involved hospitals.

Among the 43 patients, 23 had a history of cardiothoracic surgery (CS-related) and 15 had never undergone open-heart surgery (CS-unrelated). For five additional patients no information was available confirming or excluding previous open-heart surgery.

In order to compare our isolates with those already reported as part of the global epidemic we included five published environmental isolates from water supply and from new built 3T-HCUs at LivaNova production site (n = 2; Subgroup 1.8; GenBank accession nos. ERR1463901, ERR1463898), 3T-HCUs in use in Switzerland (n = 2; Subgroup 1.1 and Branch 1; GenBank accession nos. ERR1464041, ERR1463965) and at Maquet (Rastatt, Germany) HCU production site (n = 1; Subgroup 1.8; GenBank Accession no. ERR1464127).

WGS was performed on the Illumina NextSeq 500 platform and the reads were mapped on *M. chimaera* DSM-44623 as reference genome (GenBank accession no. LQOO00000000) using Burrows-Wheeler Aligner [5]. All datasets reached a mean coverage >50 fold, with at least 80% of the reference genomes positions complying with the thresholds of variant detection (minimum depth of coverage of 10x and 75% allele frequency). Variant calling was done using the widely employed programs Samtools [6] and the Genome Analysis Toolkit (GATK) [7].Custom perl scripts were used to filter the variants with a minimum coverage of 10 reads in both forward and reverse orientation, 10 reads calling the allele with a phred score of ≥20, and 75% allele frequency. The combined set of detected high quality SNP positions was used to construct a Maximum parsimony tree using RAxML version 8 [8] with a general time reversible substitution model, 1,000 re-samples and Gamma20 likelihood optimization to account for rate heterogeneity among sites. The resulting phylogenetic tree was then visualized and annotated using the online program GrapeTree [9].

In order to detect the mixed populations in the samples we reduced the threshold of variant detection (at least 2 reads calling the allele with a phred score of ≥20 and 5% allele frequency).

Average Nucleotide Identity (ANI) were calculated based on OrthoANIu algorithm [10], using *M. chimaera* DSM-44623 as reference genome. Clustal W/X software were used for multiple sequence alignment of the sequences [11] and subsequent calculation of phylogenetic networks and visualization were done with SplitsTree V4.16.1 [12]

Single genomes were located to Groups, Subgroups and Branches in the basis of signature SNPs described previously [4]. Non-Group 1 and mixed genomes with the major subpopulation < 75% were excluded from phylogenetic analysis (n = 19 genomes).

## Results and discussion

Out of 122 genomes only three belonged to groups other than Group 1. Their distribution among the different groupings is as reported in Table 1.

All *M. chimaera* isolates from CS-related patients fitted Group 1; of these, 19 belonged to Subgroup 1.1, three to Subgroup 1.8 and one to Branch 1. Subgroups 1.1 and 1.8 were also the most frequent among the HCU isolate with 44 and 7 isolates, respectively. Moreover from 10 HCUs a mixed population of *M. chimaera* from Subgroups 1.1 and 1.8 were isolated (Tables 1 and S1). WGS analysis confirmed nine isolates from HCUs and one from a CS-related patient, all belonging to Subgroup 1.1, were identical to one strain isolated from HCU in Switzerland [4]. Furthermore, four of our isolates belonging to Subgroup 1.8, isolated from HCUs (n = 2) and CS-related patients (n = 2), were closely related (3–4 SNPs different) to the strains grown from water supply and 3T-HCU at LivaNova production site [3].

A common source of *M. chimaera* infection has been recognized on the basis of the remarkable similarity between almost all the isolates from patients with a history of cardiac surgery and the ones recovered from most HCUs [4,13]. Three distinct strains of *M. chimaera*, belonging to Subgroups 1.1, 1.8, and 2.1, have been reported responsible of contamination of LivaNova HCUs at the production site [4]. This finding is in agreement with our results showing large prevalence of Subgroups 1.1 and 1.8 among the isolates from CS-related patients and HCUs (Fig 1). Only in one CS-associated patient we isolated *M. chimaera* belonging to Branch 1; this strain was isolated from a sputum sample and the patient did not present any of signs or symptoms related to deep infection by *M. chimaera*.

The isolates from CS-unrelated patients were distributed across the phylogenetic tree, mostly belonging to Branch 1. Overall, the isolates within Subgroup 1.1 showed comparatively little diversity, with a median pairwise distance of only 4 SNPs (range 0–20). These results are in agreement with clonal M. chimaera isolates, described in HCU water samples and CS-related patient from Australia, New Zealand and also US patient strains [14,15]. Unaccountably, two isolates from two CS-unrelated patients, in different hospitals, grouped with Subgroup 1.1 and were genetically very similar to those from CS-related patients and HCUs (mean pairwise distance of <5 SNPs).

A very peculiar *M. chimaera* strain was isolated from lower respiratory infection in a CS-unrelated patient. The INNO-LiPA MYCOBACTERIA v.2 line-probe assay ascribed the strain

**Table 1. Group/subgroup distribution of *M. chimaera* isolates in the present study.**

|  | HCU | CS-related patients | CS-unrelated patients | CS-unknown patients |
|---|---|---|---|---|
| Subgroup 1.1 | 44 | 19 | 2 | 2 |
| Branch 1 | 7 | 1 | 8 | 3 |
| Subgroup 1.8 | 7 | 3 |  |  |
| Branch 2 |  |  | 2 |  |
| Group 1. ungrouped |  |  | 2 |  |
| Group 2 | 2 |  |  |  |
| Group 5* |  |  | 1 |  |
| Subgroups 1.1+1.8 | 10 |  |  |  |
| Subgroup 1.1+Branch 1 | 4 |  |  |  |
| Other Mixes | 5 |  |  |  |
| Total | 79 | 23 | 15 | 5 |

The group/subgroups were assigned based on the specific SNP signatures [4].

*Found in this study

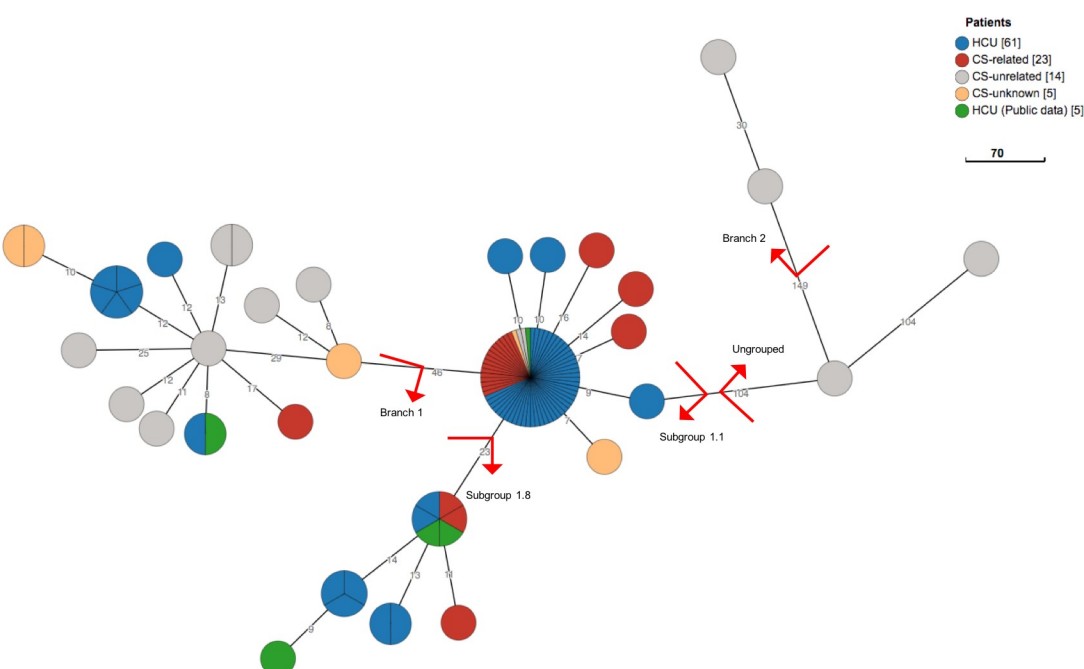

**Fig 1. Maximum parsimony tree built from 348 SNP positions of the 108 group1 isolates mapped to the genome of *M. chimaera* DSM44623 in logarithmic scale.** The *M. chimaera* genomes from HCUs, patients and Published genome data are indicated in different colours. The two major outbreak subgroups 1.1, 1.8 and also other subgroups are indicated by red labels. A cut-off of 5 SNPs was used to collapse the branches. For this analysis, we combined all group 1 isolates from our study with five other published genomes from LivaNova HCUs, water supply at production site and Maquet HCU production site. We excluded isolates for which a mixed strain population was detected based on signature SNPs [4].

to the *Mycobacterium avium complex*. Sequencing of the full 16S rRNA gene and the 16S–23S ITS identified the bacterium as *M. chimaera*. The value of ANI (98,53% identity) definitively confirmed that our strain and *M. chimaera* DSM-44623 belonged to the same species. Alignment of concatenated SNPs of our isolate and those of the four known *M. chimaera* groups performed by using clustal W/X software and subsequent visualization with SplitsTree software confirmed the phylogenetic position of all the genomes within the *Mycobacterium avium complex*, most closely related to *M. chimaera*. Interestingly, using the suggested threshold of 1000 SNPs for defining *M. chimaera* groups [4], the strain proved different from all four known groups, with the groups 1, 2, 3 and 4 being 23793, 6837, 9431 and 23818 SNPs distant respectively (Fig 2).

Present study has some limitations related to the partial coverage of the Italian *M. chimaera* epidemic and to lack of epidemiological data linking individual patients to specific HCUs.

## Conclusions

In conclusion, our data are consistent with the hypothesis that disseminated *M. chimaera* in CS patient are related with the use of a specific HCU; more the similarity of isolates in different part of the world pinpoint a common, single source of infection.

Clinicians should monitor patients who have had cardiac surgery using HCUs for signs and symptoms of *M. chimaera* infection to enable early diagnosis and treatment. Finally, we shown that WGS is the preferred method to distinguish whether a clinical strain is related to the HCU outbreak strain.

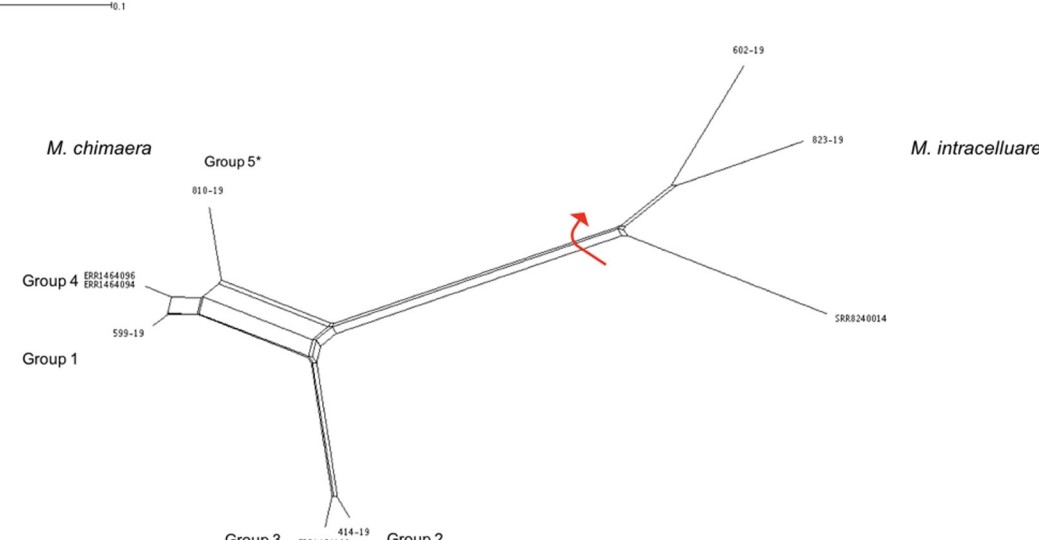

**Fig 2. NeighborNet splitstree tree.** NeighborNet splitstree tree built from 60570 SNP positions of representative *M. chimaera* genomes belonging to Groups 1, 2, 3 and 4, supporting the presence of a previously unreported grouping we named Group 5.

All sequence reads were submitted to the NCBI sequence read archive with Project number **PRJNA592124** (S1 Table).

## Supporting information

**S1 Table.**
(XLSX)

## Author Contributions

**Data curation:** Arash Ghodousi, Riccardo Manganelli, Enrico Tortoli.

**Formal analysis:** Vincenzina Monzillo.

**Investigation:** Arash Ghodousi, Marta Peracchi, Giorgio Palù, Paola Mantegani.

**Methodology:** Arash Ghodousi, Emanuele Borroni.

**Resources:** Vinicio Manfrin.

**Supervision:** Daniela Maria Cirillo.

**Validation:** Loredana Fallico.

**Writing – original draft:** Arash Ghodousi.

**Writing – review & editing:** Arash Ghodousi, Emanuele Borroni, Marta Peracchi, Giorgio Palù, Loredana Fallico, Mario Rassu, Vinicio Manfrin, Vincenzina Monzillo, Riccardo Manganelli, Enrico Tortoli, Daniela Maria Cirillo.

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
