## [Decision Letter · Decision Letter 0]

17 Jul 2020

PONE-D-20-09309

Genomic Analysis of Cardiac Surgery-Associated Mycobacterium chimaera Infections in Italy

PLOS ONE

Dear Dr. Cirillo,

Thank you for submitting your manuscript to PLOS ONE. After careful consideration, we feel that it has merit but does not fully meet PLOS ONE’s publication criteria as it currently stands. Therefore, we invite you to submit a revised version of the manuscript that addresses the points raised during the review process.

My apologies for the delayed response. More than one reviewer accepted the assignment but did not return a review. The person who completed a review had a very favorable impression of the manuscript and I agree based on my own review that it is a strong manuscript that requires only some minor modifications and clarifications. In addition to addressing the reviewer's comments below, please see my own review comments here:

Line 64: Please clarify whether "10-fold coverage" means coverage by at least 10 reads or something else.Line 64: Please provide more detail of the settings used for variant calling in samtools and GATK.Line 69: Typo: should be "GrapeTree"Line 70: Minimum 2-fold (or read?) coverage seems like it would identify many sequencing errors as false-positive variants. What steps were taken to ensure this was not the case?Line 88: The term "very similar" should be more specifically defined or quantified.Line 100: Should define the abbreviations used in the figure (i.e. "P. data") in the figure legend for clarity.Line 113: Describe how ANI was determinedLine 114: Clustal X and SplitsTree4 should be referenced and discussed in the materials and methods section.Line 131: A supplemental table listing the isolates sequenced, their group assignments, and SRA accession numbers should be added.Line 166: The PLoS One editorial office will be following up with you re: further clarification of the Ethics Statement.

Thank you for your patience in these difficult and unusual times.

We look forward to receiving your revised manuscript.

Kind regards,

Egon Anderson Ozer, MD PhD

Academic Editor

PLOS ONE

Journal Requirements:

2. In your ethics statement in the Methods and online submission information, please ensure that you have specified whether patient samples are de-identified or anonymized before access.

3. Your ethics statement must appear in the Methods section of your manuscript. If your ethics statement is written in any section besides the Methods, please move it to the Methods section and delete it from any other section. Please also ensure that your ethics statement is included in your manuscript, as the ethics section of your online submission will not be published alongside your manuscript.

Reviewers' comments:

Reviewer's Responses to Questions

**Comments to the Author**

1. Is the manuscript technically sound, and do the data support the conclusions?

Reviewer #1: Yes

2. Has the statistical analysis been performed appropriately and rigorously? 

Reviewer #1: No

3. Have the authors made all data underlying the findings in their manuscript fully available?

Reviewer #1: Yes

4. Is the manuscript presented in an intelligible fashion and written in standard English?

Reviewer #1: Yes

5. Review Comments to the Author

Reviewer #1: Excellent and concise analysis of the M.chimaera associated with HCU in the authors' country (Italy). The statistical details of the analyses were a bit low. Why use the type strain DSM 44623T versus ZUERICH1 or CDC 2015-22-71 as references for the mapping of the reads? The application of published SNP thresholds is appropriate, but the distribution and depiction of the SNP variation observed in HCU isolates from Italy is absent. Percentages of HCU clone versus non-related genotypes were absent from the text as well. Just a few more statistical numbers included in the report would be appropriate. A table or supplementary table of isolates, name, origin, and SRA or biosample number should be included in Supplementary materials for readers. Authors omitted citing the two HCU genomic analyses from the US CDC/National Jewish Health and the genomic analyses of Australian/New Zealand isolates.

6. PLOS authors have the option to publish the peer review history of their article (what does this mean?). If published, this will include your full peer review and any attached files.

Reviewer #1: No

---

## [Author Response · Author response to Decision Letter 0]

28 Aug 2020

Response to Editor and Reviewers:

Editor# Line 64: Please clarify whether "10-fold coverage" means coverage by at least 10 reads or something else. 

Thanks for the comment. We have re-wrote the phrases to clarify “10-fold coverage”.

Line 64: Please provide more detail of the settings used for variant calling in samtools and GATK. 

Thanks for the comment. More details of the setting for variant calling were added to the manuscript. 

Line 69: Typo: should be "GrapeTree"

It is now corrected.

Line 70: Minimum 2-fold (or read?) coverage seems like it would identify many sequencing errors as false-positive variants. What steps were taken to ensure this was not the case?

We’d like to thank for this comment. In this case we have used the parameters in which the majority allele other than wild type is considered per position. For each isolate, we calculated the mean allele frequency for each set of SNP alleles, setting the following thresholds: minimum mapping quality of 20, minimum base quality at a position of 20, minimum read depth at a position of 2X, maximum strand bias for a position 90% in order to distinguish the real genomic variants from possible sequencing errors.

Line 88: The term "very similar" should be more specifically defined or quantified.

Thanks for the comment, the mentioned term was defined more specifically in the text.

Line 100: Should define the abbreviations used in the figure (i.e. "P. data") in the figure legend for clarity.

The figure legend has been corrected.

Line 113: Describe how ANI was determined 

The calculation of ANI has been described in Materials & Methods section.

Line 114: Clustal X and SplitsTree4 should be referenced and discussed in the materials and methods section. 

Thanks for the comment. Clustal X and SplitsTree4 have been referenced and discussed in the materials and methods section.

Line 131: A supplemental table listing the isolates sequenced, their group assignments, and SRA accession numbers should be added. 

Thank you for this comment. A supplementary table listing the isolates sequenced, their group assignment, and SRA accession numbers has been added and submitted with the revised manuscript.

Line 166: The PLoS One editorial office will be following up with you re: further clarification of the Ethics Statement.

Reviewer #1: Excellent and concise analysis of the M.chimaera associated with HCU in the authors' country (Italy). The statistical details of the analyses were a bit low. Why use the type strain DSM 44623T versus ZUERICH1 or CDC 2015-22-71 as references for the mapping of the reads? The application of published SNP thresholds is appropriate, but the distribution and depiction of the SNP variation observed in HCU isolates from Italy is absent. Percentages of HCU clone versus non-related genotypes were absent from the text as well. Just a few more statistical numbers included in the report would be appropriate. A table or supplementary table of isolates, name, origin, and SRA or biosample number should be included in Supplementary materials for readers. Authors omitted citing the two HCU genomic analyses from the US CDC/National Jewish Health and the genomic analyses of Australian/New Zealand isolates.

Response:

We would like to thank the reviewer for the comment. In order to analyze the WGS data we have used the M. chimaera DSM 44623 and also M. chimaera strain Zuerich1 as reference genome and we didn’t find significant differences for phylogenetic analysis. However, genome size of M. chimaera DSM 44623 and strain Zuerich1 are different (5,865,644 vs 6,175,731 bp, respectively). For genotyping we extracted a set of variants specific for (sub)groups of isolates defined by the phylogenetic analysis with respect to the M. chimaera DSM 44623 genome and its annotation with reference to van Ingen et al. Lancet Infect Dis 2017;17: 1033–41.

Moreover, a more detailed statistic of HCU and clinical isolates was included the result section.

A supplementary table of isolates, name, origin, and SRA accession number has been included in Supplementary materials. 

Finally, the two mentioned papers were cited in the manuscript and included in the references.

---

## [Decision Letter · Decision Letter 1]

3 Sep 2020

Genomic Analysis of Cardiac Surgery-Associated Mycobacterium chimaera Infections in Italy

PONE-D-20-09309R1

Dear Dr. Cirillo,

We’re pleased to inform you that your manuscript has been judged scientifically suitable for publication and will be formally accepted for publication once it meets all outstanding technical requirements.

Kind regards,

Egon Anderson Ozer, MD PhD

Academic Editor

PLOS ONE

Additional Editor Comments (optional):

Reviewers' comments:

Reviewer's Responses to Questions

**Comments to the Author**

1. If the authors have adequately addressed your comments raised in a previous round of review and you feel that this manuscript is now acceptable for publication, you may indicate that here to bypass the “Comments to the Author” section, enter your conflict of interest statement in the “Confidential to Editor” section, and submit your "Accept" recommendation.

Reviewer #1: All comments have been addressed

2. Is the manuscript technically sound, and do the data support the conclusions?

Reviewer #1: Yes

3. Has the statistical analysis been performed appropriately and rigorously? 

Reviewer #1: Yes

4. Have the authors made all data underlying the findings in their manuscript fully available?

Reviewer #1: Yes

5. Is the manuscript presented in an intelligible fashion and written in standard English?

Reviewer #1: Yes

6. Review Comments to the Author

Reviewer #1: Thank you for an excellent revised manuscript draft. All of my suggestions and points of review have been addressed.

7. PLOS authors have the option to publish the peer review history of their article (what does this mean?). If published, this will include your full peer review and any attached files.

Reviewer #1: No

---

## [Editor Report · Acceptance letter]

16 Sep 2020

PONE-D-20-09309R1 

Genomic Analysis of Cardiac Surgery-Associated *Mycobacterium chimaera* Infections in Italy 

Dear Dr. Cirillo:

I'm pleased to inform you that your manuscript has been deemed suitable for publication in PLOS ONE. Congratulations! Your manuscript is now with our production department. 

Kind regards, 

on behalf of

Dr. Egon Anderson Ozer 

Academic Editor

PLOS ONE